# Relationships between Physical Activity Frequency and Self-Perceived Health, Self-Reported Depression, and Depressive Symptoms in Spanish Older Adults with Diabetes: A Cross-Sectional Study

**DOI:** 10.3390/ijerph20042857

**Published:** 2023-02-06

**Authors:** Angel Denche-Zamorano, Jorge Perez-Gomez, Sabina Barrios-Fernandez, Rafael Oliveira, Jose C. Adsuar, João Paulo Brito

**Affiliations:** 1Promoting a Healthy Society Research Group (PHeSO), Faculty of Sport Sciences, University of Extremadura, 10003 Cáceres, Spain; 2Health, Economy, Motricity and Education (HEME) Research Group, Faculty of Sport Sciences, University of Extremadura, 10003 Cáceres, Spain; 3Occupation, Participation, Sustainability and Quality of Life (Ability Research Group), Nursing and Occupational Therapy College, University of Extremadura, 10003 Cáceres, Spain; 4Sports Science School of Rio Maior, Polytechnic Institute of Santarém, 2040-413 Rio Maior, Portugal; 5Research Center in Sport Sciences, Health Sciences and Human Development, Quinta de Prados, Edifício Ciências de Desporto, 5001-801 Vila Real, Portugal; 6Life Quality Research Centre, 2040-413 Rio Maior, Portugal

**Keywords:** physical exercise, physical fitness, mental health, quality of life, noncommunicable diseases, sedentary lifestyles, health

## Abstract

Diabetes is one of the most prevalent noncommunicable diseases in the world. This disease can affect both physical and mental health in the population. This study analyzed the prevalence of Self-Perceived Health (SPH), self-reported depression, and depressive symptoms in comparison with the Physical Activity Frequency (PAF) reported by Spanish older adults with diabetes. A cross-sectional study was carried out with data from 2799 self-reported diabetic participants, all of whom were residents of Spain, aged 50–79 years, and included in the European Health Surveys carried out in Spain (EHIS) both in 2014 and 2020. The relationships between the variables were analysed with a chi-squared test. A z-test for independent proportions was performed to analyze differences in proportions between the sexes. A multiple binary logistic regression was carried out on the prevalence of depression. Linear regressions were performed on depressive symptoms and SPH. Dependent relationships were found between the SPH, self-reported depression, and depressive symptoms with PAF. Most of the very active participants reported a higher prevalence of self-reported depression. Physical inactivity increased the risk of depression, major depressive symptoms, and negative SPH.

## 1. Introduction

Diabetes prevalence has increased over the last decades, becoming a major public health problem [1]. The prevalence of diabetes in adults worldwide is 451 million people, and it is estimated to increase to 693 million by 2045 [2]. Moreover, while around 5 million deaths worldwide are attributable to the disease, and about 850 billion dollars in global health expenditure can be attributed to this condition, an estimated 50% of people with diabetes may be undiagnosed. Complications associated with diabetes involve microvascular (most commonly neuropathy) and macrovascular diseases affecting several organs including muscle, skin, heart, brain, and kidneys [3]. The experience of living with diabetes is often associated with disease-specific concerns and can cause conditions such as diabetes distress, psychological insulin resistance, and the persistent fear of hypoglycemic episodes [4].

Traditionally, management of this condition has been focused on the control of glycaemic and physical problems, but as scientific evidence about mental issues in people with diabetes, and health sciences generally, have become more holistic [5], the psychological well-being of the diabetic population has become essential. Research has shown a clear relationship between diabetes and a variety of mental health issues [6], including depressive diseases [7,8,9,10,11,12,13,14]. Evidence suggests that the consequences of diabetes and depression co-occurrence (mortality, costs, severity of illness) are worse than when these conditions happen separately [15,16]. Hence, as diabetes is related to mental health problems, interventions involving mental health professionals can be beneficial for both conditions, although findings are inconclusive and suggest an increased risk of early all-cause mortality [17]. For this reason, alternatives for managing diabetes-related health and mental health problems should be explored [6].

A primary goal in diabetes intervention is to promote Health-Related Quality of Life (HRQoL). Thus, considering peoples’ perspectives is essential to assess intervention outcomes [18], and therefore, patient self-reported outcomes must be used [19]. Self-Perceived Health (SPH) refers to a person’s overall perception of their physical and mental status [20]. This is useful, as it provides summarized information about the respondent’s objective and subjective health perception [21]. It is a reliable status predictor since it integrates the objective condition knowledge and the individual’s understanding of his or her physical and mental symptomatology [22]. 

Physical activity (PA) is defined as any bodily movement produced by skeletal muscles that result in energy expenditure [23]. Recommended PA types for people with diabetes include aerobic exercise, strength, endurance, flexibility, and balance training [24]. PA confers positive effects in terms of physical (immune system, blood lipid profile, blood pressure, cardiovascular disease, endothelial function, and physical fitness) [25,26,27] and mental health [28,29]. Consequently, PA could be a positive lifestyle modification with multiple health benefits for the disease [6,28,30,31]. For all the above reasons, this study aimed to analyze the relationships between SPH, self-reported depression, and depressive symptoms with Physical Activity Frequency (PAF) in Spanish older adults with diabetes.

## 2. Materials and Methods

### 2.1. Design and Ethical Concerns

We conducted a cross-sectional study with data from two European Health Surveys that were carried out in Spain in 2014 (EHIS2014) [32] and 2020 (EHIS2020) [32]. These surveys were carried out by trained and accredited personnel from the Spanish National Institute of Statistics in collaboration with the Spanish Ministry of Health, Consumption and Social Welfare; the interviews were conducted between January 2014 to February 2015 (EHIS2014) [32] and July 2019 to July 2020 (EHIS2020) [33]. The EHIS is conducted among the Spanish adult population (over 15 years of age) and aims to determine the population’s health status and indicators and sociodemographic factors. Therefore, standardized questionnaires are used to compare the responses within European countries to evaluate and plan health-related actions. This study followed Commission Regulation N° 141/2013 implementing Regulation 1338/2008 of the European Parliament and the Council on Community statistics on public health and health and safety at work, as regards statistics based on the EHIS.

### 2.2. Participants

As the EHIS sampling system and sample calculation describe [32,33], we used a three-stage randomized automatic sampling system with stratification, for which the first-stage units were census sections. In the second stage, household dwellings were randomly selected from these census sections. In the third stage, one adult per household was randomly selected. The sampling system and sample calculation are described in the methodologies of both surveys. Inclusion criteria for this research included (1) age between 50 and 79 years and (2) having self-declared diabetes in the surveys. As a result, the following participants were excluded: 4531 individuals>80 years of age, 19,854 individuals of <50 years of age, 17,704 non-diabetics, and 26 participants with no diabetes information. For analyses that included other variables, such as body mass index (BMI), depression status, and depressive symptoms, participants who did not submit data on the questions for these variables were not considered.

In the EHSS2014, in those over 80 years of age with diabetes, only 4 individuals (0.9%) reported a frequency of physical activity between several times a month and several times a week. No statistical inference was possible with such a small sample. This was similar to the EHSS 2020 sample, where only 25 individuals with diabetes and over 80 years of age performed physical activity several times a month or week, between 1 and 2%. For this reason, those participants were not included in the sample.

The total number of individuals with self-reported diabetes analyzed was 2799 (1319 in the EHISS2014 and 1480 in the EHISS2020). The median ages were significantly higher among women than among men (69 years vs. 66 years in EHISS2014 and 69 years vs. 68 years in the EHISS2020). Figure 1 presents the participant selection process. 

### 2.3. Procedures

Body mass index (BMI) was calculated through weight/height^2^ (kg/m^2^) and was used to split the participants into the following groups: Underweight; Normal; Overweight and Obesity [34].

Self-Perceived Health (SPH) data were extracted from the Q.21 item (“In the last 12 months, would you say that your health status has been very good, good, fair, poor, very poor?”), with three levels of possible answers: Positive (answers: good; very good); Fair (answers: fair), and Negative (answers: poor; very poor).

Depression status data were extracted from the Q.25a.20 item (“I will read you a list of diseases or health problems. Do you have or have you ever had depression?”), with four possible answers: yes; no; don’t know; no answer.

Depressive symptoms data were extracted from variable depressive symptomatology. This variable could take values between 0 and 24, with 0 being the lowest symptomatology and 24 the highest, as it was obtained with the sum of the eight-item Patient Health Questionnaire Depression scale (PHQ-8) scores [35] from the Q.41.a-Q.41.h items in both surveys. The PHQ-8 is a reliable questionnaire validated in the Spanish population to detect depressive symptoms [36]. According to the PHQ-8 scores, in the EHIS2014-EHIS2020 surveys [32,33], the following symptomatologic levels were established: none (less than 5 points), mild (between 5 and 9 points), moderate (between 10 and 14 points), moderately severe (between 15 and 19 points), and severe (more than 19 points). In this study, none = none in EHIS; minor = mild and moderate in EHIS; severe = moderately severe and severe in EHIS.

Physical Activity Frequency (PAF) data extracted were from the 112 items in both surveys (“which of these possibilities describes better the frequency in which you perform some PA in your free time?”), with possible answers being “I don’t exercise. I spend my free time almost entirely sedentary: reading, watching TV, going to the cinema, etc.”, considered as inactive; “I practice some PA or sport occasionally (walking or cycling, gentle gymnastics, recreational activities involving light exertion, etc.”), considered as occasional; “I do PA several times a month: sports, gymnastics, jogging, swimming, cycling, team games, etc.”, considered as active; “I do sport or physical training several times a week”, considered as very active; don’t know/no answer. Answer options for these questions were never, occasional, several/month, and several/week, respectively. Participants who answered don’t know or no answer were excluded.

Diabetes status data were extracted from item Q.25a.12 (“I will read you a list of diseases or health problems. Do you have or have you ever had Diabetes?”), with possible answers being yes, no, don’t know, or no answer.

### 2.4. Statistical Analysis

A Kolgomorov–Smirnov test was performed to analyze data distribution followed by the study variables. Continuous variables (age and BMI) were presented with median and interquartile range (IQR), analyzing possible differences between sexes with the Mann–Whitney U test. The remaining (categorical) variables BMI group, PAF, depression status, depressive symptoms, and SPH were presented by their absolute and relative frequencies, analyzing possible differences in proportions between sexes with a z-test for independent proportions. Associations between categorical variables were analyzed with the chi-squared test, calculating the contingency coefficient to assess the strength of the association [37]. A multiple binary logistic regression was carried out, considering depression status as the dependent variable and sex, age, BMI, and PAF as independent variables. Two multiple linear regressions were performed, taking SPH and depressive symptoms as dependent variables and sex, age, BMI, and PAF as independent. A significance level of less than 0.05 was assumed in all analyses. The IBM SPSS Statistics v.25 for Windows (IBM Corp., Armonk, NY, USA) software was used.

## 3. Results

Table 1 shows the descriptive analysis of the older Spanish population with self-reported diabetes, derived from the EHIS2014 [32]. More than 80% presented as overweight (43%) or obesity (37%). Men had a higher prevalence of being overweight than women (49% vs. 36%, *p* < 0.05), while women had a higher prevalence of obesity (43% vs. 32%, *p* < 0.05). There was a relationship between sex and BMI groups (*p* < 0.001). Moreover, 93% were considered inactive (47%) or occasional (46%) according to their PAF. The inactive women prevalence was higher than men (39% vs. 56%, *p* < 0.05), with a dependent relationship between the FPA and sex (*p* < 0.001). The self-reported depression prevalence was 22%, being higher in women than in men (31% vs. 14%, *p* < 0.05). Associations between sex and depression prevalence were found (*p* < 0.001), with similar results in depressive symptoms according to the PHQ-8 questionnaire. Depressive symptoms were associated with sex (*p* < 0.001), and men’s prevalence without depressive symptoms were higher than women’s (81% vs. 58%, *p* < 0.05). Men showed a higher SPH prevalence compared with women (49% vs. 58%, *p* < 0.05), showing associations between SPH and sex (*p <* 0.05). As there were many response options, almost all test values were significant; therefore, the probability of error is quite low. Although significance is high, the contingency coefficient is, in some cases, weak to medium.

The EHIS2020 showed that 77% were overweight (44%) or obese (33%). There were more men with overweight than women (48% vs. 40%, *p <* 0.05), and there was more obesity among women than men (36% vs. 30%, *p <* 0.05). There were also associations between the BMI group and sex (*p =* 0.005). Further, 87% of the sample presented a frequency of inactive (43%) or occasional (44%) PA, with a higher prevalence of inactive women than men (48% vs. 39%, *p <* 0.05). Associations were found between FPA and sex (*p =* 0.002). The prevalence of self-reported depression was 18%, higher in women than in men (25% vs. 11%, *p <* 0.05). There were also associations between sex and self-reported depression (*p <* 0.001). These same associations were found between sex and depressive symptoms, according to PHQ-8 (*p <* 0.001). A higher proportion of men without depressive symptoms was found compared to women (83% vs. 71%, *p <* 0.05). Finally, men had a higher prevalence of positive health than women (51% vs. 22%, *p <* 0.05), with significant associations between sex and SPH (Table 2).

Table 3 shows the associations between PAF and depression prevalence, depressive symptoms, and SPH among 50–79-year-old Spanish patients with diabetes from both the EHIS2014 and EHIS2020. Both surveys found associations between PAF and Self-Reported Depression, depressive symptoms, and SPH prevalence (*p <* 0.001).

Figure 2 shows the self-reported depression prevalence according to PAF. In both surveys, self-reported depression prevalence was much higher in inactive people (28% and 24%, respectively) than in the rest of the PAF groups. The lowest prevalence was found in those groups that performed PA several times per week (8%) in EHIS2014 and occasionally or several times per week (13%) in EHIS2020.

The highest prevalence of minor or severe symptoms of depression was found in the inactive groups, both in the ENSE2014 (32.0% and 11.3%, respectively) and in the EHIS2020 (26% and 6%, respectively) (Appendix A). Figure 3 shows the proportions of people without depressive symptoms, according to the PAF. The highest proportions of people without symptoms were found in the groups with the highest frequency of physical activity; several times a week: 94% (EHIS2014) and 87% (EHIS2020).

Multiple binary regression models on depression explained the 8.5% figure for EHIS2014 and 7.9% for EHIS2020 (Nagelkerke R^2^, 0.2 to 0.4). Inactive people and women were the ones who presented the highest risk of self-reported depression in the EHIS2014 and EHIS2020 (Table 4). 

Figure 4 shows how the highest negative SPH prevalence was found in the inactive groups (36% in the EHIS2014 vs. 28% in the EHIS2020), being lower than in the rest of the PAF groups.

The prevalence of positive SPH was higher in those groups with higher PAF (68% in EHIS2014 vs. 56% in EHIS2020) as shown in Figure 5.

The Wald test in the multiple binary regression model for self-reported depression risk factors found that all explanatory variables in the model were significant. Appendix A shows the linear regression models on depressive symptoms. The coefficient of determination (R^2^) was 10% for EHIS2014 vs. 3.8% for EHIS2020. Appendix A shows the linear regression models on the SPH; the coefficient of determination (R^2^) was 10.6% for EHIS2014 vs. 5.5% for EHIS2020.

## 4. Discussion

This study aimed to analyze the associations between mental health and SPH with PAF in Spanish older adults with self-reported diabetes. The strength of this analysis comes from the sample selection method—multi-stage, stratified random sampling—which resulted in a representative sample concerning multiple demographic characteristics, including diabetes presence. The self-reported diabetes prevalence observed in this study was 5% for EHIS2014 and 6.7% for EHIS2020 (Figure 1), which is not in line with the figures declared by the International Diabetes Federation (8.8%) [38].

### 4.1. Association between Sex and Body Mass Index (Overweight and Obesity)

There was a relationship between the sex and BMI groups both in the EHIS2014 and the EHIS2020. Moreover, men had a higher overweight prevalence while women had a higher obesity prevalence (Table 2). The reason for this is unclear. It may be associated with factors such as differences in the distribution of adipose tissue or fat mass for a given BMI level [39,40,41]. However, according to the Abdullah meta-analysis [42], obesity was associated with a risk of diabetes risk that was seven times greater compared to normal weight, while being overweight was associated with a three-fold greater risk. The same authors also stated that women tended to report slightly higher relative risks compared to men. The diabetes-relative risk for women with obesity was eight times higher compared to women with normal weight, while men with obesity had a risk that was six times higher compared to men with normal weight. Similar differences in obesity–diabetes relative risk between women and men were also found in another meta-analysis [43]. Given the relevance of biology, interpretation may be more focused on sex rather than sex differences (where sex differences could be defined as sexual-social and psychological differences between men and women), although it is often difficult to separate their effects one from another [44,45]. 

### 4.2. Physical Activity Frequency

From the present analysis perspective, a relationship was also found between sex and PAF. Several studies to date have reported the inverse association between PA with type 2 diabetes (T2DM) in middle-aged populations [46,47,48,49]. Interestingly, in the Canadian National Population Health Survey, 1996–1997 [50], exercise was not significantly associated with diabetes in men. However, physical inactivity was considered a T2DM predictor in both sexes [21]. A PA decrease has been often described in the elderly (>60 y) [51,52,53,54,55]). In this study, both in the EHIS2014 and the EHIS2020, the men’s median age (66 y and 68 y) was lower than in women (69 y and 69 y) with equal statistical dispersion (IQR = 12) (Table 1 and Table 2). This difference may account for the higher prevalence of inactive women.

The Di@bet.es Study is a national population-based survey conducted in Spain during 2009–2010 to examine diabetes prevalence including physical inactivity [56]. Based on the population of the Di@bet.es study, PA was evaluated through the SF-IPAQ in 4991 individuals (median age 50 years, 57% women) [57]. Low PA was present in 44% of individuals with known diabetes, in line with the current study. Sedentariness prevalence was 32.3% for men and 39% for women, with notable sex differences at early and older ages. Related to old age, less healthy individuals tended to be physically inactive, even those diabetic individuals at high risk for cardiovascular disease [57]. Educational level was linked to a healthier lifestyle and therefore associated with lower diabetes prevalence [58,59].

Analysis carried out across age groups showed that diabetes was associated with a decrease in physical functioning and PAF, with a higher prevalence in women [56]. PA levels in women with diabetes were also lower than in their male counterparts, as based on National Health and Nutrition Examination Survey (NHANES) data and baseline data from large interventional studies [60,61]. PAF disparities between sexes in adults with diabetes were exacerbated in populations with lower levels of education, but sex disparity persisted at all levels [62].

### 4.3. Self-Reported Depression Prevalence

Self-reported depression prevalence was reported in EHIS2014 (Table 1) and EHIS2020 (Table 2), being higher in women compared to men. Associations between sex and depression prevalence were found. Moreover, associations with depressive symptoms, according to PHQ-8, were also found in both surveys: the prevalence of men without depressive symptoms was higher than women (Table 1 and Table 2). The present findings are in opposition to others which have suggested that there were no significant association between known diabetes and depressive symptoms, such as one study on women in a German community after controlling for co-morbidities [63]. However, other studies reported significantly higher depressive symptoms prevalence in diabetic women than in men [13,64,65]. It is unclear whether prevalence rates differ according to the diabetes type [66].

In both EHIS2014 and EHIS2020, individuals with self-reported depression had a higher lack of PAF compared to individuals without depression (Table 1 and Table 2). Furthermore, individuals who did not report depressive symptoms presented a higher PAF (several times a week, a month, and occasionally). However, the percentage of individuals who did not present depressive symptoms and who did not perform PA was high in both surveys. There is a strong relationship between depression and depressive symptoms and low PAF in the Spanish diabetic population aged 50–79 years in the EHIS2014-2020 (Table 4). Our findings are consistent with the existing literature on other populations [67,68,69,70,71,72,73,74,75].

It has been reported that depression prevalence increases among the elderly with chronic medical conditions like diabetes [71,74]. Hence, while the diabetes prevalence increased, self-reported depression prevalence decreased in EHIS2014 and EHIS2020. This last survey revealed a value (17.7%) like the 17% reported by Pawaskar et al. [74] with a south-eastern United States population. Two meta-analyses [7,76] on depression and depressive symptoms in T2DM reported a wider-ranging prevalence of 10.9–32.9%. The depression prevalence rates in people with diabetes were significantly higher—at least double for those with diabetes compared to those without any chronic disease [7,77,78,79]. In contrast, a few studies in Europe (UK and Germany) and Canada reported lower depression prevalence rates in people with T2DM [63,80,81].

Depression and depressive symptoms in T2DM are associated with adverse diabetes-related outcomes, including problems in self-management, poor glycemic control, increased risk of diabetes complications, and higher mortality. The increased risk of depression was associated with lower HRQoL and higher impairments in activities of daily living, particularly in the elderly [68,74]. Diabetes management is complex, and depressive symptoms and depression may be associated with different diabetes manifestations such as temporary episodes of hypoglycemia or hyperglycemia [68,71,72]. The specific mechanisms that explain this association are still not fully understood due to the high variability among patients regarding the effects of diabetes on mood and self-care behaviors [79]. The influencing mechanisms found in the literature point to physiological mechanisms such as the effect of blood sugar level on mood [7,9].

Green et al. [68] found that self-reported hypoglycemia was more prevalent among individuals with T2DM and was associated with lower HRQoL and a greater burden of depression. Diabetes is associated with a higher risk of death and being diagnosed as diabetic often has a strong impact on patients beyond the physiological effects of the disease [7]. Most people tend to experience anxiety and depression after their diagnosis because of what it means to them and the uncertainty about their future and the feeling of loss of health due to the diagnosis [10].

### 4.4. Self-Perceived Health

Factors related to the quality of life and the health conditions of the diabetic person could be understood through the assessment of positive Self-Perception of Health [82]. The perspective of diabetic adults about their SPH and how it relates to PA could be understood through their association with social and health determinants.

Self-Perceived Health is considered to be a wide-ranging and rapid metric. Therefore, it has been proposed for evaluating people’s health conditions due to the relationship between perceived and actual health status [83].

In the EHIS2014 and the EHIS2020, men presented a higher prevalence of positive SPH than women, with significant associations between sex and SPH (Table 1 and Table 2). Additionally, the prevalence of positive SPH was higher in those groups with higher PA (Table 3). It should be emphasized that subjective health assessment considers not only the state of somatic but also psychic health [21]. Moreover, it reflects the capability of individuals to function in each social and organizational environment. Subjective assessment of health has been found to highly correlate with the results of its objective assessment and health status indices [84,85,86].

Recently, much interest has been focused on the impact of PA on the modification of subjective health assessment in adults [87]. The present study has pointed to a strong association between insufficient PA and lower self-perceived health status. However, it was reported that in healthy individuals, there is a small beneficial main effect of PA on subjective well-being, independent of the prior fitness level of the participants and various characteristics of the PA intervention [88]. Furthermore, a large heterogeneity of studies published on this relation warrants further research regarding underlying mechanisms.

Despite diabetes and/or other chronic disorders playing an important role in SPH, age may have a significant impact on men and women regardless of their comorbidities [89]. Moreover, there were evident differences in older diabetics’ SPH rating their health as poor to fair compared to non-diabetics. Geographic location can also instances differences in patients with self-reported diabetes HRQoL [19]. Another potential reason for health disparities among people with T2DM could be sex differences in healthcare management and socio-environmental factors [90,91,92], including age, level of education, mode of treatment, and treatment adherence, and PA is a significant factor for this.

### 4.5. Limitations

There are some limitations in the current study due to the self-reported nature of the data on diabetes diagnosis. The diagnosis is usually confirmed with gold-standard parameters such as the homeostasis model assessment of insulin resistance (HOMA-IR), fasting blood glucose level, blood HbA1c level, or using data from medical records or National Health Fund registries, which was not the case here. However, several studies have stated that self-reported diabetes may be characterized by moderate sensitivity (55–80%), high specificity (84–97%), and reasonably good positive and reliable predictive values (>92%) over time [19,93,94,95].

In this study, there was a lack of central obesity measures, and both body weight and height were not objectively measured, but self-reported. There is a tendency for men to overreport their height and women to under-report it. Such reporting bias is small and non-differential to disease outcomes, having little impact on association estimation [96]. In addition, the duration and type of diabetes were not asked, which could strengthen a useful explanatory analysis. The questionnaire also has some limitations. It does not use precise tools like accelerometers in the PA assessment [97]. The PA measure was self-reported and retrospective, which may have led to recall bias and to it being more prone to validity and reliability issues [98]. Two important variables to characterize PAF, namely, intensity and duration, were not included in the survey. Furthermore, PAF overreporting by those interviewed must be considered; the sedentariness prevalence could be even higher [57,99,100].

The study questionnaire did not contemplate specific health conditions that could hamper PA practice and that are frequent in T2DM and/or obese individuals, which could explain, in part, the levels of sedentary lifestyles. Another limitation was the lack of socio-economic characterization and educational level of the sample, which are usually linked to a healthier lifestyle and therefore can be factors that promote a better health perception. Finally, our analysis included an interval age range between age 50 and 79 years, which could influence the results once it has been reported to be an SPH determinant [89]. Finally, depression determinants may also vary according to the pharmacotherapeutic class of antidiabetic medications, but such factors were not controlled in this study [74,76].

Future analyses should include other socioeconomic variables, such as educational level, occupation, number of children in households, and lifestyle measures such as smoking, alcohol consumption, diet, and other mental health variables, such as fatigue and stress.

## 5. Conclusions

The hypothesis of this study was confirmed (Table 5).

Diabetic adults’ self-perception refers to the correlation between health condition and functionality and seems to be a good indicator of the quality of life, morbidity, and functional decline. Diabetes appears to increase the risk of developing depression and depressive symptoms. Therefore, early detection and treatment intervention provides the best protective mechanisms available against the effects of depression on diabetes outcomes, and a psychological service provision for people with diabetes is needed.

PAF is inversely related to depression and depressive symptoms incidence, which highlights the importance of PA among lifestyle interventions designed to prevent diabetes. Therefore, we urge healthcare providers to consider PAF when counselling diabetic patients. Our data can assist in providing healthcare to adults with diabetes and can also contribute to direct intersectoral actions that can positively and longitudinally affect this population’s well-being.

## Figures and Tables

**Figure 1 ijerph-20-02857-f001:**
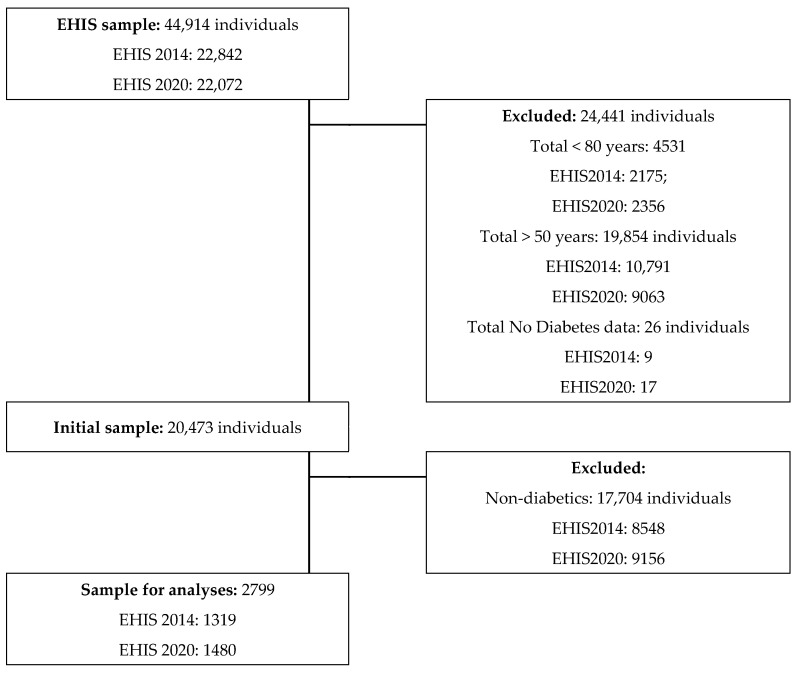
Participants flow chart.

**Figure 2 ijerph-20-02857-f002:**
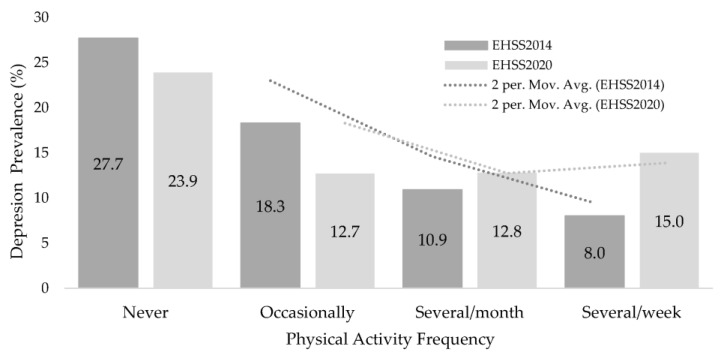
Depression prevalence according to the Physical Activity Frequency in 50–79-year-old Spanish individuals with diabetes from the EHIS2014 and EHIS2020.

**Figure 3 ijerph-20-02857-f003:**
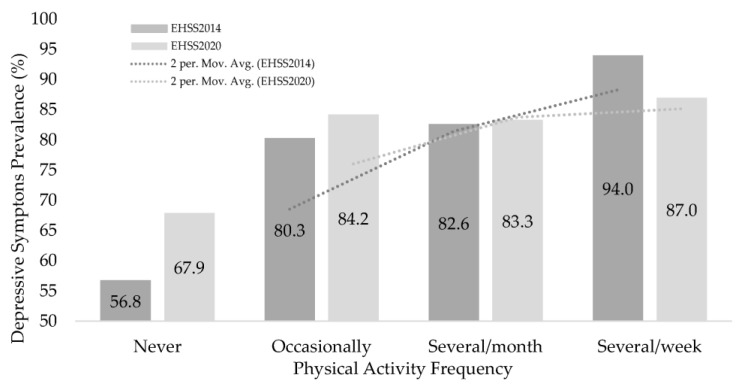
People without depressive symptoms according to the Physical Activity Frequency in EHIS2014 and EHIS2020.

**Figure 4 ijerph-20-02857-f004:**
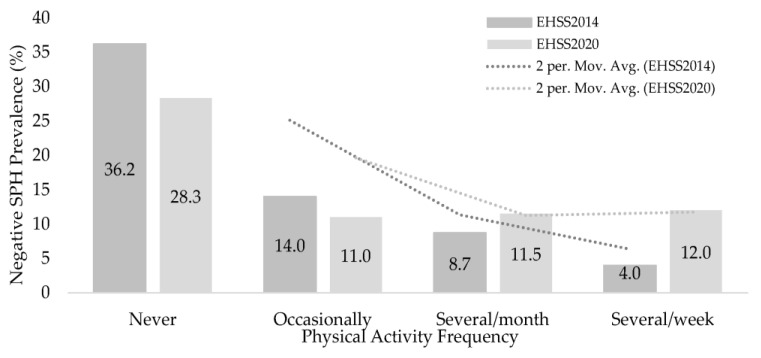
Negative self-perceived health prevalence according to the Physical Activity Frequency.

**Figure 5 ijerph-20-02857-f005:**
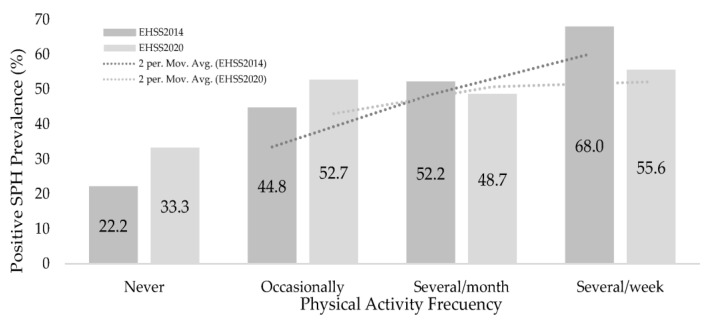
Positive self-perceived health prevalence, according to the Physical Activity Frequency.

**Table 1 ijerph-20-02857-t001:** Descriptive analysis: age, BMI groups, Physical Activity Frequency, depression, depressive symptoms, and Self-Perceived Health in the 50–79 years Spanish population with diabetes, according to EHIS2014.

	Overall (*n* = 1319)		Men (*n* = 687)		Women (*n* = 632)					
Variables	Mdn	IQR	Mdn	IQR	Mdn	IQR	x^2^	df	*p*	CC
Age	67	12	66	12	69	12	n.a.	n.a.	<0.001	n.a.
BMI	28.4	6.4	28	5.6	29	7.4	n.a.	n.a.	0.003	n.a.
**BMI Group**	*n*	%	*n*	%	*n*	%			*p* *	
Underweight	6	0.5%	2	0.3%	4	0.7%	21.8	3	<0.001	0.132
Normal	240	19.4%	125	18.7%	115	20.2%				
Overweight	534	43.1%	327	48.8%	207	36.3% *				
Obesity	460	37.1%	216	32.2%	244	42.8% *				
**Physical Activity Frequency**	*n*	%	*n*	%	*n*	%			*p* *	
Inactive	621	47.1%	269	39.2%	352	55.7% *	42.2	3	<0.001	0.176
Occasional	601	45.6%	349	50.8%	252	39.9% *				
Active	46	3.5%	33	4.8%	13	2.1% *				
Very active	50	3.8%	35	5.1%	15	2.4% *				
**Depression**	*n*	%	*n*	%	*n*	%			*p* *	
Yes	291	22.1%	94	13.7%	197	31.2% *	58.8	1	<0.001	0.207
No	1027	77.9%	593	86.3%	434	68.8% *				
**Depressive symptoms**	*n*	%	*n*	%	*n*	%			*p* *	
None	915	69.9%	551	80.7%	364	58.1% *	80.5	2	<0.001	0.241
Minor	306	23.4%	108	15.8%	198	31.6% *				
Severe	88	6.7%	24	3.5%	64	10.2% *				
**Self-Perceived Health**	*n*	%	*n*	%	*n*	%			*p* *	
Positive	465	35.3%	288	41.9%	177	28% *	36.1	2	<0.001	0.163
Fair	539	40.9%	272	39.6%	267	42.2%				
Negative	315	23.9%	127	18.5%	188	29.7% *				

*n*, participants; Mdn, median; IQR, interquartile range; x^2^, Pearson chi-squared; df, degrees of freedom; *p*, *p*-value from chi-squared test; CC, contingency coefficient; n.a., not applicable; *, significant differences between sex ratios. *p <* 0.05 from z-test; BMI, body mass index; EHIS, European Health Survey Spain.

**Table 2 ijerph-20-02857-t002:** Descriptive analysis: age, BMI group, Physical Activity Frequency, depression, depressive symptoms, and Self-Perceived Health in the 50–79 years population with diabetes, according to EHIS2020.

Overall (*n* = 1480)			Men (*n* = 799)		Women (*n* = 681)					
Variables	Mdn	IQR	Mdn	IQR	Mdn	IQR	x^2^	df	*p*	CC
Age	68	12	68	12	69	12	n.a.	n.a.	0.003	n.a
BMI	27.9	6.0	27.7	5.4	28.1	6.8	n.a.	n.a.	0.060	n.a.
**BMI Group**	*n*	%	*n*	%	*n*	%	x^2^	df	*p* *	CC
Underweight	3	0.2%	0	0.0%	3	0.5%	13.0	3	0.005	0.096
Normal	321	22.9%	176	22.7%	145	23%				
Overweight	622	44.3%	369	47.7%	253	40.2% *				
Obesity	458	32.6%	229	29.6%	229	36.3% *				
**Physical Activity Frequency**	*n*	%	*n*	%	*n*	%	x^2^	df	*p* *	CC
Inactive	637	43%	310	38.8%	327	48% *	14.4	3	0.002	0.096
Occasional	657	44.4%	384	48.1%	273	40.1% *				
Active	78	5.3%	40	5%	38	5.6%				
Very active	108	7.3%	65	8.1%	43	6.3%				
**Depression**	*n*	%	*n*	%	*n*	%	x^2^	df	*p* *	CC
Yes	261	17.7%	89	11.2%	172	25.3% *	50.7	1	<0.001	0.182
No	1216	82.3%	709 + 229	88.8%	507	74.7% *				
**Depressive symptoms**	*n*	%	*n*	%	*n*	%	x^2^	df	*p* *	CC
None	1137	77.4%	659	83.1%	478	70.7% *	34.3	2	<0.001	0.151
Minor	280	19.1%	118	14.9%	162	24% *				
Severe	52	3.5%	16	2%	36	5.3% *				
**Self-Perceived Health**	*n*	%	*n*	%	*n*	%	x^2^	df	*p* *	CC
Positive	656	44.3%	406	50.8%	250	21.7% *	30.1	2	<0.001	0.141
Fair	550	47.2%	267	33.4%	283	41.6%				
Negative	274	18.5%	126	15.8%	148	29.7% *				

*n*, participants; Mdn, median; IQR, interquartile range; x^2^, Pearson chi-squared; df, degrees of freedom; *p*, *p*-value from chi-squared test; CC, contingency coefficient; n.a., not applicable; *, significant differences between sex ratios. *p <* 0.05 from z-test; BMI, body mass index; EHIS, European Health Survey Spain.

**Table 3 ijerph-20-02857-t003:** Depression, depressive symptoms, and Self-Perceived Health according to the Physical Activity Frequency in the 50–79 years population with diabetes, according to EHIS2014 and EHIS2020.

EHIS2014												
	Never		Occasionally		Several Times/Month		Several Times/Week					
**Depression**	*n*	(%)	*n*	(%)	*n*	(%)	*n*	(%)	x^2^	df	*p*	CC
Yes	172	(27.7%)	110	(18.3%)	5	(10.9%)	4	(8%)	25.6	3	<0.001	0.138
No	448	(72.3%)	491	(81.7%)	41	(89.1%)	46	(92%)				
**Depressive symptoms**	*n*	(%)	*n*	(%)	*n*	(%)	*n*	(%)	x^2^	df	*p*	CC
None	348	(56.8%)	481	(80.3%)	38	(82.6%)	47	(94%)	104.6	6	<0.001	0.272
Minor	196	(32%)	99	(16.5%)	8	(17.4%)	3	(6%)				
Severe	69	(11.3%)	19	(3.2%)	0	(0%)	0	(0%)				
**Self-Perceived Health**	*n*	(%)	*n*	(%)	*n*	(%)	*n*	(%)	x^2^	df	*p*	CC
Negative	225	(36.2%)	84	(14.0%)	4	(8.7%)	2	(4%)	143.4	6	<0.001	0.313
Fair	258	(41.5%)	248	(41.3%)	31	(39.1%)	35	(28%)				
Positive	138	(22.2%)	269	(44.8%)	24	(52.2%)	34	(68%)				
**EHISS2020**												
	**Never**		**Occasionally**		**Several times/month**		**Several times/week**					
**Depression**	*n*	(%)	*n*	(%)	*n*	(%)	*n*	(%)	x^2^	df	*p*	CC
Yes	152	(23.9%)	83	(12.7%)	10	(12.8%)	16	(15%)	30.1	3	<0.001	0.141
No	484	(76.1%)	573	(87.3%)	68	(87.2%)	91	(85%)				
**Depressive symptoms**	*n*	(%)	*n*	(%)	*n*	(%)	*n*	(%)	x^2^	df	*p*	CC
None	428	(67.9%)	550	(84.2%)	65	(83.3%)	94	(87%)	60.9	6	<0.001	0.200
Minor	164	(26%)	92	(14.1%)	12	(15.4%)	12	(11.1%)				
Severe	38	(6.0%)	11	(1.7%)	1	(1.3%)	2	(1.9%)				
**Self-Perceived Health**	*n*	(%)	*n*	(%)	*n*	(%)		*n* (%)	x^2^	df	*p*	CC
Negative	180	(28.3%)	72	(11%)	9	(11.5%)	13	(12%)	89.9	6	<0.001	0.239
Fair	245	(38.5%)	239	(36.4%)	31	(39.7%)	35	(32.4%)				
Positive	212	(33.3%)	346	(52.7%)	38	(48.7%)	60	(55.6%)				

*n*, participants; x^2^, Pearson chi-squared; df, degrees of freedom; *p*, *p*-value from chi-squared test; CC, contingency coefficient; EHIS, European Health Survey Spain.

**Table 4 ijerph-20-02857-t004:** Multiple binary regression model for self-reported depression risk factors.

EHIS 2014								
	β	S.E.	Wald	df.	Sig.	Exp(β)	95% C.I. for EXP(β)	
							Lower	Upper
PAF (Never)			9.901	3	0.019			
Occasionally	−0.322	0.149	4.662	1	0.031	0.724	0.541	0.971
Several/month	−0.796	0.494	2.594	1	0.107	0.451	0.171	1.189
Several/week	−1.172	0.538	4.755	1	0.029	0.310	0.108	0.888
Sex (Women)	0.980	0.149	42.954	1	0.000	2.663	1.987	3.569
Age	−0.008	0.009	0.733	1	0.392	0.992	0.975	1.010
BMI	0.022	0.014	2.352	1	0.125	1.022	0.994	1.051
Constant	−1768.000	0.769	5.280	1	0.022	0.171		
**EHIS 2020**								
	**β**	**S.E.**	**Wald**	**df.**	**Sig.**	**Exp(β)**	**95% C.I. for EXP(β)**	
							**Lower**	**Upper**
PAF (Never)			18.481	3	0.000			
Occasionally	−0.646	0.158	16.791	1	0.000	0.524	0.385	0.714
Several/month	−0.696	0.359	3.762	1	0.052	0.498	0.247	1.007
Several/week	−0.411	0.294	1.944	1	0.163	0.663	0.372	1.181
Sex (Women)	0.905	0.148	37.282	1	0.000	2.473	1.849	3.307
Age	−0.014	0.009	2.152	1	0.142	0.987	0.969	1.005
BMI	0.023	0.014	2.603	1	0.107	1.023	0.995	1.053
Constant	−1.465	0.788	3.458	1	0.063	0.231		

β, understandarized beta; S.E., standard error of regression; Wald, Wald chi-squared test; df, degrees of freedom; Sig, statistical significance; Exp., exponential regression; C.I., confidence interval; PAF, Physical Activity Frequency; BMI, body mass index; EHIS, European Health Survey Spain.

**Table 5 ijerph-20-02857-t005:** Confirmation of the hypothesis of the study.

Hypothesis	Confirmation
Prevalence of overweight or obesity both from the EHIS2014 and EHIS2020	confirmed
Relationship between sex and BMI groups both from the EHIS2014 and EHIS2020	confirmed
Prevalence of inactive or occasional physical activity frequency both from the EHIS2014 and EHIS2020	confirmed
Associations between sex and depression and depressive symptoms prevalence both from the EHIS2014 and EHIS2020	confirmed
Associations between Self-Perceived Health and sex both from the EHIS2014 and EHIS2020	confirmed

## Data Availability

Microdata were obtained on the website of the Spanish Ministry of Health, Consumer Affairs, and Social Welfare: https://www.sanidad.gob.es/estadisticas/microdatos.do (accessed on 2 March 2022).

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
