# Peer review of "Relationships between Physical Activity Frequency and Self-Perceived Health, Self-Reported Depression, and Depressive Symptoms in Spanish Older Adults with Diabetes: A Cross-Sectional Study"

_ijerph, 2023, doi:10.3390/ijerph20042857_

Round 1
Reviewer 1 Report
Congratulations - the article is very interesting and important to the field of public health as it relates to diabetics. I emphasize the large amount of literature used in the introduction and discussion. The article is well structured and the statistical analyses are very comprehensive. The use of the EHIS dataset in such modeling is unique. The sample of more than 2,000 diabetic patients is sufficiently representative.
I will suggest to add some minor changes:
1. Please rational why you remove people aged >80 years
2. Discussion: please refer to tables, figure in previous chapters for instance: “men had a higher overweight prevalence while women had a higher obesity prevalence” –which table? Please made correction in all subsection in Discussion.
3. Summary of all hypotheses confirmed in one table (rows and columns with variables and information about confirmed/not confirmed relationship)
4. Please add some guidelines for further research. Development of panel study? EHIS wave 2 and wave 3 combined. Inclusion of different socio-economic variables such as occupation or number of kids , or lifestyle measures such as smoking, alcohol , diet (number of vegetables/fruits portions) or mental health variables such as fatigue etc. Such data are included in EHIS dataset.
Author Response
Dear Reviewer,
Please see the attachment.
Thank you.
KR

Reviewer 2 Report
Hello - thanks for the opportunity to review this article. It represents an interesting study on a very important topic - consideration of the issues of " Self-Perceived Health, Depression and Physical Activity”
Please see the detailed comments below for specific issues:
Abstract: abstract is clear.
Introduction is clear and concise
Materials and Methods:
Although the sample is described in Figure 1 (109), some elements should be present in the text to improve comprehension: sample size, age distribution (mean, standard deviation), gender (mean, standard deviation), etc.
Particularly striking is the absence of the educational level of the participants. In the discussion (296) it appears as a relevant data to explain the relationship between healthier lifestyle and Diabetes prevalence. Is it possible to have this data in the sample?
Discussion: Well described and supported by other research, but lacking explanatory elements, including conjecture about possible explanations for the data (e.g. explaining the association between depression and diabetes).
On several occasions, the data already presented in the results section (305-310; 322,) are mentioned again, which is redundant and increases the feeling of descriptiveness to the detriment of explanations.
Author Response

(The authors gave the same response as above.)
